# Association between high-fasting insulin levels and metabolic syndrome in non-diabetic middle-aged and elderly populations: a community-based study in Taiwan

Yun-Hung Chen,[1] Yu-Chien Lee,[1] Yu-Chung Tsao,[1,2,3] Mei-Chun Lu,[1] Hai-Hua Chuang,[4] Wei-Chung Yeh,[1,3] I-Shiang Tzeng,[5] Jau-Yuan Chen[1,3]

[1]Department of Family Medicine, Chang Gung Memorial Hospital, Linkou Branch, Taoyuan, Taiwan
[2]Department of Occupational Medicine, Chang Gung Memorial Hospital, Linkou Branch, Taoyuan, Taiwan
[3]Chang Gung University College of Medicine, Taoyuan, Taiwan
[4]Department of Family Medicine, Chang Gung Memorial Hospital, Taipei Branch, Taipei, Taiwan
[5]Department of Research, Taipei Tzu Chi General Hospital, BuddhistTzu Chi Medical Foundation, New Taipei City, Taiwan

**Correspondence to**
Dr Jau-Yuan Chen;
welins@cgmh.org.tw

## ABSTRACT

**Objectives** We aimed to determine the association between fasting insulin (FI) levels and metabolic syndrome (MetS) in non-diabetic middle-aged and elderly adults in a community in Taiwan.

**Design** Cross-sectional observational study.

**Setting** Community-based investigation in Guishan township of northern Taiwan.

**Participants** Our study included adults aged 50 years and above during community health examinations between January and October 2014. People with diabetes mellitus were excluded. A total of 321 people were enrolled.

**Outcome measures** We divided participants according to tertiles of FI as low, medium and high levels. Pearson correlation was assessed between insulin level and each of the diagnostic components of metabolic syndrome (MetS-DCs) with adjustment of age. The prevalence of MetS-DCs based on tertiles of FI were studied and analysed by Cochran–Armitage trend test. The risk for prevalence of MetS in the middle and high insulin group as compared with the low insulin group were assessed by multivariate logistic regression with adjustments for age, gender, smoking, body mass index (BMI), hypertension and hyperlipidaemia. Youden Index was performed for the optimised cut-off value.

**Results** Our results showed positive correlation of FI level with systolic blood pressure, waist circumference, fasting plasma glucose and triglyceride levels, while negative correlation was shown with high-density lipoprotein (P<0.001). The prevalence of each MetS-DCs increased as a trend while FI levels increased (P<0.001). OR (95% CI) of MetS was 5.04 (2.15 to 11.81) for high insulin groups compared with the low insulin group after adjusting confounders (P<0.001). Area under receiver operating characteristic curve (ROC) curve (AUC) was 0.78, and cut-off value 7.35 µU/mL for FI was obtained (sensitivity: 0.69; specificity: 0.77).

**Conclusions** Middle-aged and elderly non-diabetic people with increased FI are associated with a higher prevalence of MetS in the community in Taiwan. Furthermore, FI is an independent risk factor of MetS in this study population.

## Strengths and limitations of this study

► This is the first study to explore the relationship between fasting insulin and MetS in the middle-aged and elderly populations of non-diabetic Asian people.
► We offer a biomarker to identify middle-aged and elder non-diabetic Taiwanese with MetS.
► We adjusted many confounding factors to make the results more reliable.
► The recall bias from self-reported lifestyle behaviours is unavoidable.
► The causal relationship between fasting insulin level and MetS cannot be demonstrated in our cross-sectional study.

## INTRODUCTION

Metabolic syndrome (MetS) is associated with a cluster of unhealthy metabolic risk factors, including abdominal obesity (excess body fat around the waist), glucose intolerance, pre-morbid hypertension and dyslipidaemia.[1–3] A number of studies have reported that MetS increases the risk of type 2 diabetes mellitus (T2DM), cardiovascular disease (CVD) and other non-communicable diseases (NCDs).[4–7] The rising prevalence of MetS has created a disturbing challenge to personal health.[8–12]

Insulin resistance has long been associated with MetS.[13–16] Basal insulin represents 45% to 50% of daily insulin,[17 18] and the FI level approximates basal insulin.[18 19] Studies have shown that FI levels are associated with the prevalence of MetS, which may be due to its representativeness of insulin resistance.[20 21] A study has even shown that elevated FI levels may predict the future incidence of MetS.[19] If insulin resistance is the foundation of MetS,[14 15] and FI represents insulin resistance with an area under curve (AUC) (95% CI)

of 0.995 (0.993 to 0.996),[20] a high FI level may be able to caution the physician for susceptibility to metabolic diseases and hence cardiovascular risks.[22] We therefore aimed to determine the association between FI levels and MetS in non-diabetic middle-aged and elderly adults in a community in Taiwan.

## METHODS

### Study participants

This was an observational and cross-sectional study conducted at Linkou Chang Gung Memorial Hospital in Taoyuan County, Taiwan between January and October 2014. The inclusion criteria included residents 50 to 90 years' old and living in Guishan township. Six hundred and nineteen residents were eligible for the study. A total of 400 residents agreed to participate in our health examination. Participants were excluded if they had diabetes. Seventy-nine participants with diabetes mellitus were excluded. Diabetes mellitus was defined as any of the following: previous diagnosis of diabetes mellitus; recent use of oral anti-hyperglycaemic drugs or insulin; or participants with fasting glucose ≥126 mg/dL. A total of 321 participants (111 males and 210 females) were ultimately enrolled for analysis. This study was approved by the Institutional Review Board of the hospital and written informed consent was obtained from all of the participants before enrollment.

### Data collection

We obtained exercise (exercising ≥3 times a week or not) and dietary habits (vegetarian or not) from self-administered questionnaires, which also included smoking (current smoker or not) and marital status (currently married or not). Anthropometric data, such as height, weight, waist circumference (WC) and blood pressure were also recorded. The participants were dressed in light clothing without shoes for weight and height measurements. The BMI was calculated as the weight in kilogrammes (kg) divided by the height in metres squared ($m^2$). Waist circumference was measured midway between the inferior margin of the lowest rib and the iliac crest in the horizontal plane while in an upright position. Systolic blood pressure (SBP) and diastolic blood pressure (DBP) were checked at least twice after 5 min of rest while seated. FI levels, lipid profile and fasting glucose were obtained by blood sampling after a 10 hours' overnight fast. Blood samples were analysed in the central laboratory of Linkou Chang Gung Memorial Hospital for fasting plasma glucose (FPG), serum total cholesterol (TC), low-density lipoprotein-cholesterol (LDL-C), high-density lipoprotein-cholesterol (HDL-C), serum triglycerides (TG) and FI levels. Serum insulin levels were determined with an ARCHITECT Insulin assay (Abbott Laboratories, IL, USA). Insulin was measured with a chemiluminescent microparticle immunessay (CMIA). The intra-assay variation and inter-assay variations were less than 2.7%. The ARCHITECT Insulin assay has a sensitivity of ≤1.0μU/ml.

### Defining MetS

MetS was defined by at least three of five metabolic syndrome diagnostic components (MetS-DCs), according to The Third Report of the National Cholesterol Education Programme Expert Panel (NCEP) on Adult Treatment Panel (ATP III) Asian diagnostic criteria.[23] The five MetS-DCs were as follows: SBP ≥130 mmHg and/or DBP ≥85 mmHg, or the use of anti-hypertensive drugs; decreased serum HDL-C concentration <40 mg/dL in men and <50 mg/dL in women, or treatment for dyslipidaemia; TG concentration ≥150 mg/dL, or on medication for hypertriglyceridaemia; hyperglycaemia: fasting plasma glucose level ≥100 mg/dL; and abdominal WC ≥90 cm in men or ≥80 cm in women.

### Statistical analysis

Participants were classified in one of three groups according to serum insulin level tertiles as the low, middle and high insulin groups. Clinical characteristics were expressed as the mean ±SD for continuous variables and number (%) for categorical variables. One-way analysis of variance (ANOVA) or a $\chi^2$ test was used to determine P-values for continuous or categorical variables, respectively. Pearson's correlation was performed for each MetS-DC in relation to FI levels. The Cochran–Armitage trend test was used to evaluate the increasing prevalence of MetS-DCs as a function of insulin level tertile. The low FI group was designated as the reference group to calculate the ORs of the prevalence of MetS in the middle and high FI groups using multivariate logistic regression. Confounded variables present as an obstacle to valid inference in MetS studies. Hypertension and dyslipidaemia are both common chronic conditions that affect a large proportion of the general adult population. Previous studies determining the association of FI and MetS also adjusted MetS-DCs.[19] Results of the adjusted model provide valid inference among MetS and insulin levels. A ROC curve was created for FI as a biomarker of MetS. The area under the ROC curve (AUC) was analysed, and the optimised cut-off point for FI, sensitivity and specificity were acquired using the maximal Youden Index. We used SPSS (version 23.0 for Windows, Taiwan), to perform the statistical analysis. Statistical significance was set at a P value <0.05.

## RESULTS

A total of 321 individuals, 111 men (34.6%) and 210 (65.4%) women, with a mean age of 63.91±8.32 years, were enrolled in this study. There were 90 study participants (28%) who met the diagnosis of MetS.

Table 1 shows the characteristics of the study population, which was divided based on the FI level in μU/mL. There were no statistically significant differences in age or gender between the low, middle and high insulin level groups, while differences did exist with respect to WC, SBP, HDL-C, TG and the proportion with MetS. Table 2 further shows the correlation between the FI level and

**Table 1** General characteristics of the study population based on insulin levels

| | | | Insulin levels | | | | |
|---|---|---|---|---|---|---|---|
| | | Low | | Middle | | High | |
| Variables | Total n=321 | n=110 | (≤4.8) | n=107 | (4.9–7.8) | n=104 | (≥7.9) | P value |
| Age (year) | 63.91 ±8.32 | 64.23 ±8.32 | | 64.47 ±8.67 | | 63.01 ±7.93 | | 0.40 |
| BMI (kg/m$^2$) | 24.36 ±3.53 | 22.41 ±3.14 | | 24.41 ±2.73 | | 26.37 ±3.54 | | <0.001 |
| Waist circumference (cm) | 84.23 ±9.51 | 79.69 ±7.57 | | 83.78 ±8.63 | | 89.51 ±9.65 | | <0.001 |
| SBP (mmHg) | 129.02 ±16.59 | 123.69 ±17.36 | | 129.52 ±14.51 | | 134.13 ±16.20 | | <0.001 |
| DBP (mmHg) | 77.01 ±10.90 | 75.43 ±11.80 | | 76.92 ±10.03 | | 78.79 ±10.60 | | 0.08 |
| ALT (U/L) | 21.74 ±11.06 | 18.94 ±7.81 | | 20.33 ±9.25 | | 26.15 ±14.05 | | <0.001 |
| Creatinine (mg/dL) | 0.76 ±0.44 | 0.69 ±0.17 | | 0.85 ±0.66 | | 0.75 ±0.34 | | 0.03 |
| FPG (mg/dL) | 89.10 ±9.93 | 85.29 ±9.11 | | 89.18 ±8.52 | | 93.05 ±10.60 | | <0.001 |
| HDL-C (mg/dL) | 55.70 ±14.05 | 60.93 ±14.85 | | 55.59 ±13.17 | | 50.28 ±11.94 | | <0.001 |
| Insulin (µU/mL) | 7.10 ±4.14 | 3.60 ±0.94 | | 6.21 ±0.86 | | 11.72 ±4.02 | | <0.001 |
| LDL-C (mg/dL) | 121.48 ±32.05 | 118.90 ±34.55 | | 126.03 ±31.01 | | 119.53 ±30.10 | | 0.20 |
| T-cholesterol (mg/dL) | 200.61 ±35.20 | 198.85 ±36.98 | | 203.81 ±35.12 | | 119.18 ±33.43 | | 0.52 |
| TG (mg/dL) | 117.34 ±60.61 | 95.39 ±45.13 | | 111.04 ±49.83 | | 147.05 ±72.48 | | <0.001 |
| Current smoking, n(%) | 34 (10.6) | 14 (12.7) | | 11 (10.3) | | 9 (8.7) | | 0.62 |
| Marital status (single), n(%) | 54 (16.8) | 22 (20.0) | | 14 (13.1) | | 18 (17.3) | | 0.39 |
| Men, n(%) | 111 (34.6) | 41 (37.3) | | 39 (36.4) | | 31 (29.8) | | 0.46 |
| Regular exercise, n(%) | 264 (82.2) | 92 (83.6) | | 96 (89.7) | | 76 (73.1) | | 0.01 |
| Vegetarian, n(%) | 20 (6.2) | 7 (6.4) | | 7 (6.5) | | 6 (5.8) | | 0.97 |
| HTN, n(%) | 150 (46.7) | 43 (39.1) | | 47 (43.9) | | 60 (57.7) | | 0.02 |
| Hyperlipidaemia, n(%) | 204 (63.6) | 58 (52.7) | | 69 (64.5) | | 77 (74.0) | | 0.005 |
| Metabolic syndrome, n(%) | 90 (28.0) | 11 (10.0) | | 23 (21.5) | | 56 (53.8) | | <0.001 |

Clinical characteristics are expressed as the mean±SD for continuous variables and n(%) for categorical variables. P values were derived from one-way analysis of variance (ANOVA) for continuous variables and $\chi^2$ test for categorical variables.
Notes, ranges of FI levels of different tertile groups are shown in brackets at the top of the table, units in µU/mL.
ALT, alanine aminotransferase; BMI, body mass index; DBP, diastolic blood pressure; FPG, fasting plasma glucose; HDL-C, high-density lipoprotein cholesterol; HTN, hypertension; LDL-C, low-density lipoprotein cholesterol; SBP, systolic blood pressure; TG, triglyceride.

all MetS-DCs, even after adjusting for age. FI was positively correlated with SBP, WC, FPG and TG, and negatively correlated with HDL-C, as shown in table 2. Table 3 shows the prevalence of MetS-DCs (hypertension, hyperglycaemia, dyslipidaemia and central obesity) according to the insulin level tertiles. The prevalence of MetS-DCs increased as the FI level increased, as shown by significant P values (Cochran–Armitage trend test). Figure 1 shows that the low insulin level group had a 10% prevalence of MetS, the middle insulin level group had a 21.5% prevalence of MetS and the high insulin level group had a 53.8% of MetS (P<0.0001 (Cochran–Armitage trend test)), suggesting that the prevalence of MetS increased with an increase in FI levels.

When designating the low insulin level group as the reference, the middle and high insulin level groups had an OR of 2.46 (P=0.02) and 10.50 (P<0.001) for MetS, respectively. After adjusting age, gender and BMI, the middle and high insulin level groups had an OR of 1.71 (P=0.20) and 5.63 (P<0.001) for MetS, respectively. After adjusting age, gender, BMI, smoking, hypertension and dyslipidaemia, the middle and high insulin level groups still had an OR of 1.51 (P=0.35) and 5.04 (P<0.001; table 4) for MetS, respectively. There was no significant difference between the middle and low tertile groups, but a significant difference between the high and low tertile groups, even after adjusting for the above risk factors. Based on this data, the high insulin level group had a five-fold risk for MetS compared with the low insulin level group.

Figure 2 shows the ROC curve of FI as a biomarker for MetS. The AUC was 0.78. The optimised cut-off value for insulin was 7.35 µU/mL, with a sensitivity of 0.69 and a specificity of 0.77.

## DISCUSSION
In this community-based study, we investigated fasting serum insulin levels in association with the prevalence of MetS in non-diabetic middle-aged and elderly Taiwanese adults. In our study, the prevalence of MetS in the relatively healthy middle age-to-elderly population was 28%, which is similar to the 29.75% findings reported by Li

**Table 2** Pearson's correlation coefficients for each component of metabolic syndrome and age in relation to insulin levels

| | Insulin(n=321) | | | |
| | Unadjusted | | Adjusted for age | |
| Variables | Pearson's coefficient | P value | Pearson's coefficient | P value |
|---|---|---|---|---|
| Age (year) | −0.04 | 0.50 | NA | NA |
| SBP (mmHg) | 0.21 | <0.001 | 0.22 | <0.001 |
| DBP (mmHg) | 0.11 | 0.05 | 0.10 | 0.07 |
| Waist circumference (cm) | 0.43 | <0.001 | 0.44 | <0.001 |
| FPG (mg/dL) | 0.38 | <0.001 | 0.39 | <0.001 |
| HDL-C (mg/dL) | −0.37 | <0.001 | −0.37 | <0.001 |
| TG (mg/dL) | 0.37 | <0.001 | 0.37 | <0.001 |

DBP, diastolic blood pressure; FPG, fasting plasma glucose; HDL-C, high-density lipoprotein cholesterol; NA, not applicable; SBP, systolic blood pressure; TG, triglyceride.

*et al.*[24] Among middle-aged and elderly populations in Taiwan. Looking at the three FI tertiles, there was a rising proportion of MetS as the FI level increased, also shown in previous studies.[20 25 26] This finding not only applied to MetS, but to MetS-DCs as well. The WC, SBP, TG and FPG levels were the lowest in the low FI group and highest in the high FI group: the converse applied to HDL and vice versa (table 1). This finding led us to speculate that an association exists between FI levels and MetS-DCs. We found a statistically significant correlation between

**Table 3** Prevalence of components of metabolic syndrome based on insulin levels

| Components | Low (n=110) n(%) | Middle (n=107) n(%) | High (n=104) n(%) | P value for Cochran–Armitage trend test |
|---|---|---|---|---|
| High blood pressure* | 56 (50.9) | 63 (58.9) | 78 (75) | 0.0003 |
| High blood glucose† | 8 (7.3) | 10 (9.3) | 25 (24.0) | 0.0004 |
| Low HDL-C‡ | 15 (13.6) | 19 (17.8) | 43 (41.3) | <0.0001 |
| High TG§ | 17 (15.5) | 27 (25.2) | 42 (40.4) | <0.0001 |
| Central obesity¶ | 34 (30.9) | 57 (53.3) | 82 (78.8) | <0.0001 |

*SBP≧130 mmHg or DBP≧85 mmHg, or self-reported hypertension
†Fasting blood glucose ≥100 mg/dL or self-reported diabetes mellitus
‡HDL-C <40 mg/dL in men or <50 mg/dL in women
§TG ≥150 mg/dL
¶Waist circumference ≥90 cm in men or ≥80 cm in women
DBP, diastolic blood pressure; HDL-C, high-density lipoprotein-cholesterol; SBP, systolic blood pressure; TG, triglyceride.

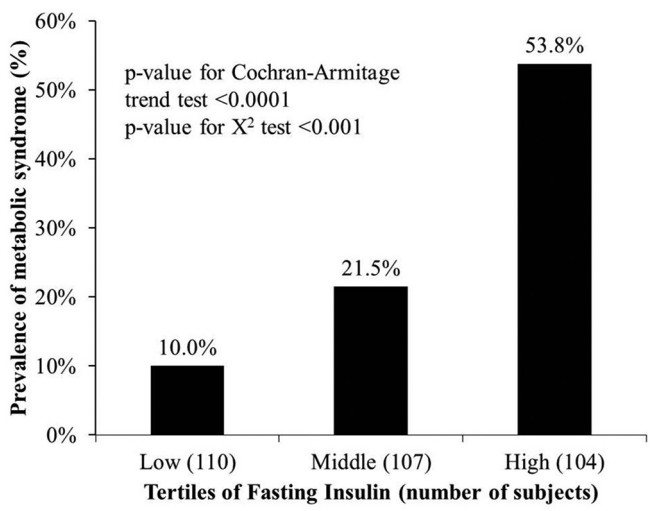

**Figure 1** Prevalence of metabolic syndrome based on insulin levels. A linear increasing trend across insulin tertiles.

FI levels and each MetS-DC, even after adjusting for age (table 2). A trend existed between the FI level and the prevalence of all five MetS-DCs (table 3). We thus wanted to know if the trend applied to the prevalence of MetS. Figure 1 shows an apparent increase in the prevalence of MetS as the FI level increased. The trend was confirmed by the Cochran–Armitage trend test (P<0.0001).

After adjusting for gender, age, BMI, smoking status, hypertension and dyslipidaemia, the middle-aged and elderly populations in the high FI group were at significant risk for developing MetS (OR=5.04, 95% CI=2.15 to 11.81; P<0.01: table 4). This conclusion is consistent with previous findings.[20 26 27] Not only is FI an independent risk factor for MetS, but in the cohort study of Sung *et al*, it was also reported that elevated FI predicted the future incidence of MetS.[19] One possible explanation might be the relationship between the FI level and insulin resistance,[20 21] which has a fundamental role in MetS.[28 29] Although the mechanism by which FI may represent insulin resistance was not investigated in the present study, a number of studies have shown that FI is a suitable surrogate marker for insulin resistance,[21 30–33] calculated by the fasting insulin resistance index (FIRI) or homeostasis model assessment of insulin resistance (HOMA-IR). A higher FI level is associated with insulin resistance in patients with impaired fasting glucose, but may be an inappropriate marker in diabetics with poor glycaemic control. It has been reported that the FI level is highly associated with MetS.[20] In our study, the AUC for FI as an indicator for MetS was 0.78, similar to another study's AUC of 0.77.[20] Based on our search of the literature, there is no widely accepted reference range for FI. A reference range for FI of 1.57 to 16.32 µU/mL has been proposed in Chinese men, but the reference range varies between different ethnicities and genders.[34] A FI level above 9µU/mL has been reported to identify 80% of patients with pre-diabetes.[35] Although we obtained a cut-off value for fasting insulin, due to large variations

**Table 4** Association between insulin levels and metabolic syndrome

| Variables | Model 1 | | | Model 2 | | | Model 3 | | |
|---|---|---|---|---|---|---|---|---|---|
| | OR | (95% CI) | P value | OR | (95% CI) | P value | OR | (95% CI) | P value |
| Low | 1.00 | – | – | 1.00 | – | – | 1.00 | – | – |
| Middle | 2.46 | (1.14 to 5.35) | 0.02 | 1.71 | (0.76 to 3.85) | 0.20 | 1.51 | (0.64 to 3.57) | 0.35 |
| High | 10.50 | (5.05 to 21.84) | <0.001 | 5.63 | (2.53 to 12.53) | <0.001 | 5.04 | (2.15 to 11.81) | <0.001 |
| P value for trend | | | <0.001 | | | <0.001 | | | <0.001 |

Model 1: unadjusted.
Model 2: adjusted for gender, age and BMI.
Model 3: adjusted for factors in model 2 plus smoking, HTN, and hyperlipidaemia.

in insulin assays, this value of >7.35 should not be generalised to other laboratory sites.

Our findings may have an impact on health screening policies in non-diabetic people older than middle-age. Elevated FI may act as an accompanying marker to enhance the risk of MetS. We do not propose to discard MetS criteria, but suggest that elevated FI may alert physicians to the risk of MetS in clinical settings of non-diabetic individuals. Given the fact that elevated FI is not only associated with a greater risk for developing MetS[19 36] but is also associated with a greater number of cardiometabolic risk factors,[22] healthy behaviour should be considered when the FI level is relatively higher in the population. We are in need of large trials to determine if participants with early stages of insulin resistance can benefit from interventions.

Some limitations in our study merit consideration. First, the principal limitation relevant to the interpretation of our results was the use of a cross-sectional design, thus a causal relationship between the FI level and MetS cannot be inferred. Second, the sample size in our study was relatively small (n=321 (the power was not calculated)) and the participants were recruited from a regional community. The participants could therefore only be distributed into three groups and the results cannot be generalised to other ethnicities. Third, although males tend to have a lower participation rate in studies,[37] there may still have been a selection bias due to the higher participation of women than men in our study. Fourth, the FI cut-off value varies between different ethnic groups and insulin assays, so physicians should be aware of this variation in clinical settings. Besides, the false negative rate (31%) should be taken into consideration when applying this data. Furthermore, even though we used a standardised questionnaire, recall and reporting bias are unavoidable for self-reported data. Finally, we did not ask participants to sleep adequately or to avoid vigorous exercise the day before blood testing, which could affect the accuracy of the fasting serum insulin level.

Our study also has strengths. First, our participants were recruited during a community health examination and represent a relatively healthy population. The effects of important confounders, including ethnicity, residential area and environmental factors, were minimised. Second, we used standardised laboratory examination protocols and anthropometric measurements. Third, while evaluating the association between MetS and the FI level, we excluded diabetic patients to avoid the effect of anti-diabetic medications on the FI level. Lastly, due to the trend of world ageing, our study aimed for middle-aged and elderly populations. Studies from all around the world indicate the relationship of FI and MetS (table 5), and our study contributes to the Taiwanese population.

In the future, we will continue to follow this community and record the development of newly diagnosed MetS. Counselling of healthy behaviours for residents with elevated FI will also be our topic of interest hereon. Whether lifestyle modification could retard the development of MetS in high FI individuals requires further studies to elaborate.

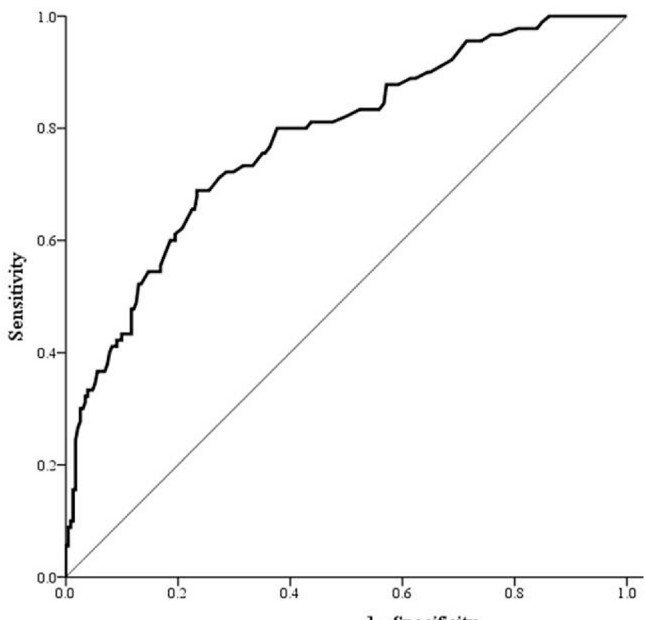

**Figure 2** ROC curve for insulin as a biomarker of metabolic syndrome. AUC 0.78. FI 7.35µU/mL (sensitivity: 0.69; specificity: 0.77).

## CONCLUSIONS

Our study provides a method to identify the risk of MetS by testing FI levels in the middle-aged and elderly non-diabetic populations. When a non-diabetic individual is

**Table 5** Studies of association between fasting insulin and metabolic syndrome

| Authors | Study year | Study population | % MetS | Fasting insulin | Risk of MetS | Main finding | Reference |
|---------|-----------|-----------------|--------|-----------------|--------------|-------------|-----------|
| Saravia G et al | 2009 to 2010 Cross-sectional | 3200 non-diabetic males in Spain | 23 | Highest tertile (≥6.13) versus lowest (≤3.80) µU/mL | OR (95% CI) 11.36 (8.65 to 15.13) for MetS | Per each 10pmol/L (1.4 uU/mL) increase in insulin, the odds for metabolic syndrome increased by 1.43 (95%CI: 1.38 to 1.49) | 20 |
| Rutter MK et al | (1991 to 1995) to (1998 to 2001) 7-year prospective | 2616 non-diabetic adults in Europe | – | 1-quintile change in fasting insulin (pmol/L) | mean (95% CI) 0.12 (0.10 to 0.15) (MetS trait score 7-year change) | Change in metabolic trait clustering was significantly associated with baseline levels and changes in fasting insulin. | 36 |
| Sung KC et al | 2003 to 2008 5-year cohort | 2350 non-MetS in Korea | 8.5 (incidence) | Highest quartile (≥8.98) versus lowest (≤6.01) IU/ml | OR (95% CI) of developing MS 5.1 (3.1 to 8.2) | The highest quartile of the insulin levels had more than a five times greater risk of developing MS compared with the participants in the lowest quartile. | 19 |
| Kanda H et al | 2000, 2001 Cross-sectional | 456 in Mongolia | 6.4 | Highest tertile (≥10.33) versus lowest (≤6.72) mmol/L | Percentage of MetS 17.1% versus 4.6% | Fasting plasma insulin is associated with MetS in farmers, but not nomads among the Mongolian population in China. | 25 |
| STOPP-T2D PSG* 2008 | 2003 Cross-sectional | 1453 eighth grade adolescents in the USA. | 9.5 | Highest quintile (≥39.1) versus lowest (≤17.0) µU/mL | OR (95% CI) 199.64 (31.29 to 1273.7) for MetS | The highest insulin quintile was almost 200 times more likely to be classified with the metabolic syndrome than participants in the lowest quintile. | 26 |
| Adam FM et al | 2005 Cross-sectional | 128 overweight/obese in Indonesia | 68.8 | Mean fasting insulin levels 15.68±7.85 versus 3.16±2.53 (uU/ml) with five components versus one component of MetS. | | There is a strong linear increase in fasting insulin levels with an increase in the number of metabolic syndrome. | 27 |

*STOPP-T2D PSG: Studies to Treat or Prevent Paediatric Type 2 Diabetes Prevention Study Group.
Mets and MS, Metabolic syndrome.

presented with a high FI level, physicians may be alerted to the risk of MetS. Our study confirms the association between FI and MetS. Further prospective research is needed to clarify the link between FI and MetS.

**Contributors** YHC and YCL were involved in writing of the manuscript. YCT, MCL, HHC and WCY conceived and supervised the study. IST provided statistical advice. JYC contributed conceived, designed and performed the experiments, collected and analyzed the data, revising it critically for important intellectual content and final approval of the version to be submitted.

**Funding** This work was supported by Chang Gung Memorial Hospital (CORPG3C0171, CORPG3C0172, CZRPG3C0053).

**Competing interests** None declared.

**Ethics approval** The study was approved by Chang Gung Medical Foundation Institutional Review Board (102-2304B), and written informed consent was given by all the participants before enrolment.

**Provenance and peer review** Not commissioned; externally peer reviewed.

**Data sharing statement** No additional data are available.

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
