## [Reviewer comments · BMJ Open]

ARTICLE DETAILS

TITLE (PROVISIONAL)	Association between high fasting insulin levels and metabolic syndrome in non-diabetic middle-aged and elderly populations: A community-based study in Taiwan
AUTHORS	Chen, Yun-Hung; Lee, Yu-Chien; Tsao, Yu-Chung; Lu, Mei-Chun; Chuang, Hai-Hua; Yeh, Wei-Chung; Tzeng, I-Shiang; Chen, Jau-Yuan

VERSION 1 – REVIEW

REVIEWER	Tao-Hsin Tung Cheng-Hsin General Hospital, Taiwan
REVIEW RETURNED	20-Mar-2017

GENERAL COMMENTS	The Authors propose the cross-sectional study to explore the relationship between high fasting insulin levels and metabolic syndrome in non-diabetic middle-aged and elderly populations. The study designs and methods used are basically appropriate, and the interpretations of the results are reasonable. However, there are several areas where the manuscript needs to be strengthened. 1. In table 4, due to hypertension and hyperlipidemia are both components of metabolic syndrome, the authors should mention the reasons for this two factors viewed as confounding factors by logistic regression in model 3.2. The false positive and false negative rate were 23% and 31% in this study. The authors should discuss this limitation.3. Please indicate the response rate of this study.4. Please consider the comparison with the other epidemiological studies in other areas using table so make clear the significance of this study.5. Some grammatical mistakes and some sentences in which meanings are not clear. Resubmission after revision of the manuscript and check by a native English speaker is recommended. Totally, I would like to congratulate the authors for the enthusiasm invested in this study. However, the manuscript does not reach the level of quality required for publication as original article without major revision in BMJ Open.
--

REVIEWER	Kasper W. ter Horst Department of Endocrinology and Metabolism, Academic Medical Center, Amsterdam, The Netherlands
REVIEW RETURNED	31-Jul-2017

GENERAL COMMENTS	Chen et al investigated the association between fasting plasma insulin levels and the prevalence of MetS in an elderly Taiwanese population. They conclude that fasting insulin provides a convenient method to predict MetS. Overall, the analysis seems appropriate to determine the association between insulin and MetS, but not all statements are strictly supported by the data. Some additional issues should be addressed. General comments:  1. It is not clear how the authors propose to implement these findings in clinical practice and/or health screening programs. What is the added benefit of measuring fasting plasma insulin (which has sub-optimal sensitivity and specificity for the diagnosis of MetS) versus assessing the previously established MetS criteria? I agree that fasting insulin levels can be a marker of insulin resistance (albeit sub-optimal) and that insulin resistance may be an independent risk factor for future cardiometabolic disease. Along these lines, the measurement of both fasting insulin levels and MetS criteria may provide patients with additional health information. However, this was not investigated. 2. The authors propose a cutoff value for fasting insulin that is a “robust and reliable predictor” for early MetS. First, fasting insulin is shown to be associated with the presence of MetS, not with the presence of early or pre-MetS. Second, on the basis of the reported data for prevalence, sensitivity, and specificity, it follows that the proposed insulin cutoff has a positive predictive value of 54% and a negative predictive value of 86%. This means that fasting insulin >7.35 mU/ml is little better than a coin flip for the diagnosis of MetS in an elderly Taiwanese population and, in my opinion, does not support the terms “robust and reliable predictor”. 3. In several sections of the manuscript, the authors state that patients with increased fasting insulin levels are at increased risk of developing MetS or that fasting insulin levels may predict subsequent MetS. These statements are not backed up by the data, which merely show an association between current MetS and fasting insulin. On the basis of the current cross-sectional data, I suggest the authors to refrain from making assumptions regarding future MetS development. (These issues may be addressed in the follow-up study, which I believe the authors are planning to perform.) 4. The authors state that patients with insulin >7.35 mU/ml should be treated with aggressive lifestyle or medical interventions, but do not present data or literature to support the health benefits of such interventions. 5. Insulin assays vary greatly across laboratories. It is unclear what assay was used to measure insulin levels. It is also unclear how the current assay would compare to assays in other laboratories and countries. The present study can be used to determine an association between insulin and MetS in this population, but conclusions regarding exact insulin cutoff values should be clearly limited to the present insulin assay/laboratory. The authors are encouraged to read up on the ADA/EASD/IFCC Insulin Standardization Work Group.
---

	Specific comments:  1. Page 2, line 5-6: association* 2. Page 2, line 52: please define AUC as area under ROC curve 3. Page 3, line 3: this cutoff may apply to your insulin assay only. 4. Page 4, lines 49 and after: participants with diabetes were excluded. How was diabetes defined? Did you perform glucose tolerance tests? Also, did you have information on other (non-MetS) comorbidities and medication use? 5. Page 4, line 52: if this was a community-based study, why were there so many more women than men enrolled? This suggests some kind of selection? 6. Page 5, line 27 and after: please describe the laboratory methods used to analyze blood samples. Especially relevant for insulin, given the great inter-laboratory variation. 7. Page 5, line 57: were patients with diabetes not excluded? So this criterion was revised? 8. Page 6, line 3 and maybe elsewhere: please use abbreviations (such as WC) consistently or not at all. 9. Page 8, top row: do the ranges in brackets indicate the fasting insulin levels in each group? Please clarify. 10. Page 13, lines 16-32: fasting insulin does not induce insulin resistance. It may be a marker of insulin resistance, under certain conditions, but usually does not explain more than ~40% of variation in direct measures of insulin resistance. 11. Page 13, lines 37 and after: range greatly depends on study population and insulin assay, and precise insulin values cannot be easily compared between laboratories and literature reports. 12. Page 15, Table 5: this table seems redundant, as the same information is also presented in the text.
--	--

VERSION 1 – AUTHOR RESPONSE

Reviewer 1

Comment#1

In table 4, due to hypertension and hyperlipidemia are both components of metabolic syndrome, the authors should mention the reasons for this two factors viewed as confounding factors by logistic regression in model 3.

Response#1:

We thank the reviewer for the comments and the reviewer's concern is well taken. We have thus addressed this concern by adding a paragraph to the "METHODS" section as follows:
 Confounded variables present as an obstacle to valid inference in MetS studies. Hypertension and dyslipidemia are both common chronic conditions that affect a large proportion of the general adult population. Previous studies determining the association of FI and MetS also adjusted MetS-DCs¹⁹. Results of the adjusted model provide valid inference among MetS and insulin levels. (Page 6-7, line 138 to 143)

Abbreviations: FI: fasting insulin; MetS: Metabolic syndrome; MetS-DCs: diagnostic components of metabolic syndrome

Comment#2

The false positive and false negative rate were 23% and 31% in this study. The authors should discuss this limitation.

Response#2:

We thank the reviewer for the comments and the reviewer's concern is well taken. We have thus addressed this concern by adding a paragraph to the "DISCUSSION" section as follows:

Fourth, according to our cutoff value, there is a false positive rate of 23% and a false negative rate of 31%. The false positive rate can be accepted due to our proposed intervention, early lifestyle modification, is less likely to do harm to the population. The false negative rate should be taken into consideration of the physician when applying this data. (Page 18, line 270-274)

Comment#3

Please indicate the response rate of this study.

Response#3:

Thanks for reminding us about no clearly statement about the response rate of this study. We have added a paragraph to the "METHODS" section as follows:

The inclusion criteria included residents 50-90 years old and living in Guishan township. 619 residents were eligible for the study. A total of 400 residents agreed to participate in our health exam. Subjects were excluded if they had diabetes. 79 participants with diabetes mellitus were excluded. A total of 321 participants (111 males and 210 females) were ultimately enrolled for analysis. (Page 5, line 91-96)

The response rate is 64.6%.

Comment#4

Please consider the comparison with the other epidemiological studies in other areas using table so make clear the significance of this study.

Response#4:

We thank for the reviewer's comments and suggestions. We have added a paragraph in "DISCUSSION". And added table 5.

Lastly, due to the trend of world aging, our study aimed for middle-aged and elderly populations. And given the differences of fasting insulin levels in different ethnic groups²², our study contributes to the Taiwanese population. Studies from all around the world indicate the relationship of fasting insulin and MetS (table 5), though few physicians have applied to their practice. (Page 19, line 284-287)

Comment#5

Some grammatical mistakes and some sentences in which meanings are not clear. Resubmission after revision of the manuscript and check by a native English speaker is recommended.

Response#5:

Thanks for your accurate comments about the issue. This is actually one of the weak points of our study, we sent the revised manuscript with English editing. We will submit the certificate in our e-mail, too.

Reviewer 2

Comment#1

It is not clear how the authors propose to implement these findings in clinical practice and/or health screening programs. What is the added benefit of measuring fasting plasma insulin (which has sub-optimal sensitivity and specificity for the diagnosis of MetS) versus assessing the previously established MetS criteria?

I agree that fasting insulin levels can be a marker of insulin resistance (albeit sub-optimal) and that insulin resistance may be an independent risk factor for future cardiometabolic disease. Along these lines, the measurement of both fasting insulin levels and MetS criteria may provide patients with additional health information. However, this was not investigated.

Response#1

Thanks for your accurate comments about this issue. We have added a paragraph to the "DISCUSSION" section as follows:

Our findings may have an impact on health screening policies in people older than middle-age. The diagnosis criteria of MetS differs according to gender, ethnicity and even age. A study shows that certain ethnic groups do not meet current criteria of MetS until they have reached a more advanced degree of insulin resistance²². Elevated FI, however, may act as a marker to alert physicians on the risk of MetS in this individual. Given the fact that elevated FI is not only associated with a greater risk for developing MetS¹⁹ but is also associated with a greater number of cardiometabolic risk factors²³, lifestyle modification should be considered when the FI level is > 7.35 $\mu\text{U}/\text{mL}$, due to the amelioration of metabolic abnormalities with diet or exercise interventions³⁸⁻⁴⁰. (Page 18, line 253-262)

Abbreviations: MetS: Metabolic syndrome; FI: fasting insulin

Comment#2

The authors propose a cutoff value for fasting insulin that is a "robust and reliable predictor" for early MetS. First, fasting insulin is shown to be associated with the presence of MetS, not with the presence of early or pre-MetS. Second, on the basis of the reported data for prevalence, sensitivity, and specificity, it follows that the proposed insulin cutoff has a positive predictive value of 54% and a negative predictive value of 86%. This means that fasting insulin >7.35 mU/ml is little better than a coin flip for the diagnosis of MetS in an elderly Taiwanese population and, in my opinion, does not support the terms "robust and reliable predictor".

Response#2

Thanks for your accurate comments about the issue. We have addressed this description about this term in the "CONCLUSION".

This study provides another convenient method to predict the existence of subsequent identify the risk of MetS by testing FI levels in the non-diabetic populations. We suggest a FI cut-off value of 7.35 $\mu\text{U}/\text{mL}$ to start lifestyle modifications in the middle-aged and elderly non-diabetic population. We believe that the cut-off value is a robust and reliable predictor can be of use to physicians, which cautions the risk of MetS. Providing medical counseling to patients with a FI level >7.35 $\mu\text{U}/\text{mL}$ should result in long-term health benefits. But further studies may be needed for this conclusion. (Page 19, line 292-298)

Comment#3

In several sections of the manuscript, the authors state that patients with increased fasting insulin levels are at increased risk of developing MetS or that fasting insulin levels may predict subsequent MetS. These statements are not backed up by the data, which merely show an association between current MetS and fasting insulin. On the basis of the current cross-sectional data, I suggest the authors to refrain from making assumptions regarding future MetS development. (These issues may be addressed in the follow-up study, which I believe the authors are planning to perform.)

Response#3

Thanks for your accurate comments about the issue. We have addressed this description about this term. We have corrected the terms throughout the manuscript. Our study does not show that fasting insulin predict subsequent MetS, it only shows an association between the two. We thank the reviewer for pointing out this mistake.

Comment#4

The authors state that patients with insulin >7.35 mU/ml should be treated with aggressive lifestyle or medical interventions, but do not present data or literature to support the health benefits of such interventions.

Response#4

Thanks for your accurate comments about the issue. This is actually one of the weak points of our study, we have added a paragraph to the "DISCUSSION" section as follows: Given the fact that elevated FI is not only associated with a greater risk for developing MetS^{19, 38} but is also associated with a greater number of cardiometabolic risk factors²³, lifestyle modification should be considered when the fasting insulin level is > 7.35 μ U/mL, due to the amelioration of metabolic abnormalities with diet or exercise interventions³⁹⁻⁴¹. (Page 18, line 257-262)

Comment#5

Insulin assays vary greatly across laboratories. It is unclear what assay was used to measure insulin levels. It also unclear how the current assay would compare to assays in other laboratories and countries. The present study can be used to determine an association between insulin and MetS in this population, but conclusions regarding exact insulin cutoff values should be clearly limited to the present insulin assay/laboratory. The authors are encouraged to read up on the ADA/EASD/IFCC Insulin Standardization Work Group.

Response#5

Thanks for reminding us about the variation of insulin assays. We have added a paragraph to the "METHOD" section as follows: Serum insulin levels were determined with an ARCHITECT Insulin assay (Abbott Laboratories, IL, USA). Insulin was measured with a chemiluminescent microparticle immunoassay (CMIA). The intra-assay variation and inter-assay variations were less than 2.7%. The ARCHITECT Insulin assay has a sensitivity of $\leq 1.0 \mu$ U/ml. (Page 5-6, line 113-117)

Comment#6

Page 2, line 5-6: association*

Response#6

We thank for the reviewer's comments and suggestions. We determined the association between fasting insulin levels and metabolic syndrome (Page 2, line 28-29)

Comment#7

Page 2, line 52: please define AUC as area under ROC curve

Response#7

We thank the reviewer's comments and suggestions. Area under ROC curve (AUC) was 0.78. (Page 2, line 23)

Comment#8

Page 3, line 3: this cutoff may apply to your insulin assay only.

Response#8

We thank for the reviewer's comments and suggestions. The cutoff may only apply to our insulin assay.

Comment#9

Page 4, lines 49 and after:

participants with diabetes were excluded. How was diabetes defined? Did you perform glucose tolerance tests? Also, did you have information on other (non-MetS) comorbidities and medication use?

Response#9

We thank for the reviewer's comments and suggestions. Diabetes mellitus was defined as any of the followings

1. Previous diagnosis of diabetes mellitus
2. Recent use of oral anti-hyperglycemic drugs or insulin
3. Participants with fasting glucose ≥ 126 mg/dl

We did not perform glucose tolerance test, this is a weak point of our study.

We did not have information on other (non-MetS) comorbidities and medication use.

Comment#10

Page 4, line 52:

if this was a community-based study, why were there so many more women than men enrolled? This suggests some kind of selection?

Response#10

We thank for the reviewer's comments and suggestions.

We have added the following in the "DISCUSSION". Third, though males tend to have a lower participation rate in studies⁴², there may still have been a selection bias due to the higher participation of women than men in our study. (Page 18, line 268-270)

Comment#11

Page 5, line 27 and after: please describe the laboratory methods used to analyze blood samples. Especially relevant for insulin, given the great inter-laboratory variation.

Response#11

We thank for the reviewer's comments and suggestions. We have added the insulin assay in the "METHODS"

Serum insulin levels were determined with an ARCHITECT Insulin assay (Abbott Laboratories, IL, USA). Insulin was measured with a chemiluminescent microparticle immunoassay (CMIA). The intra-assay variation and inter-assay variations were less than 2.7%. The ARCHITECT Insulin assay has a sensitivity of $\leq 1.0 \mu\text{U/ml}$. (Page 5-6, line 113-117)

Comment#12

Page 5, line 57:

were patients with diabetes not excluded? So this criterion was revised?

Response#12

We thank for the reviewer's comments and suggestions. We have corrected this error. 4) hyperglycemia: fasting plasma glucose level \geq 100 mg/dl. (Page 6, line 125,126)

Comment#13

Page 6, line 3 and maybe elsewhere:
please use abbreviations (such as WC) consistently or not at all.

Response#13

We thank for the reviewer's comments and suggestions. We have used consistent abbreviations.

Comment#14

Page 8, top row:
do the ranges in brackets indicate the fasting insulin levels in each group? Please clarify.

Response#14

We thank for the reviewer's comments and suggestions. We have added a line for clarification below table 1.

Ranges of FI levels of different tertile groups are shown in brackets at the top of the table, units in μ U/mL. (Page 9, line 168-169)

Comment#15

Page 13, lines 16-32:

fasting insulin does not induce insulin resistance. It may be a marker of insulin resistance, under certain conditions, but usually does not explain more than ~40% of variation in direct measures of insulin resistance.

Response#15

We thank for the reviewer's comments and suggestions.

Although the mechanism by which fasting insulin may represent insulin resistance was not investigated in the present study, a number of studies have shown that fasting insulin is a suitable surrogate marker for insulin resistance^{21 31-34} (Page 14, line 227-230)

Comment#16

Page 13, lines 37 and after:

range greatly depends on study population and insulin assay, and precise insulin values cannot be easily compared between laboratories and literature reports.

Response#16

We thank for the reviewer's comments and suggestions. We have added our insulin assay in the "METHODS" area and also added a new table 5 for comparing fasting insulin levels from other countries.

Comment#17

Page 15, Table 5:

this table seems redundant, as the same information is also presented in the text.

Response#17

We thank for the reviewer's comments and suggestions. We have removed Table 5 from the manuscript.

VERSION 2 – REVIEW

REVIEWER	Tao-Hsin Tung Cheng-Hsin General Hospital, Taipei, Taiwan
REVIEW RETURNED	28-Sep-2017

GENERAL COMMENTS	I am now pleased to accept this manuscript with the revised version.
--

REVIEWER	Kasper ter Horst Academic Medical Center, Amsterdam, The Netherlands
REVIEW RETURNED	28-Sep-2017

GENERAL COMMENTS	The authors state that their objective is to determine the association between fasting insulin and MetS. Their results demonstrate that these factors are in fact associated in an elderly Taiwanese population, and their methods seem appropriate to determine this association. Their primary objective seems nicely met. My major concern with the manuscript is that not all discussions and conclusions are justified by the results and the authors' conclusions go beyond their primary objective. Some of my previous comments were not addressed adequately: - "A study shows that certain ethnic groups do not meet current criteria of MetS until they have reached a more advanced degree of insulin resistance²²". Are the authors proposing to stop measuring MetS criteria in clinical practice, but instead to measure fasting insulin, or how do they vision this clinically? Their data demonstrate an association between fasting insulin and established MetS criteria; this does not demonstrate that fasting insulin can be used as an alternative for certain ethnic groups that do not meet formal MetS criteria.- The authors show that fasting insulin >7.35 has some predictive power, but the positive and negative predictive values do not support the term "robust and reliable predictor", in my opinion.- The authors recommend lifestyle interventions for those who have fasting insulin >7.35, but - to my knowledge - there is no evidence to support this. I agree, however, that we are in need of large trials to determine if subjects with early stages of insulin resistance can benefit from interventions.- I think it should be made clear that any reported insulin cutoff value can only be applied to the insulin assay that was used. Due to large variation in insulin assays, the value of >7.35 should not be generalized to other laboratory sites. Minor: - Their definition of diabetes should be included with the exclusion criteria/methods.
--

VERSION 2 – AUTHOR RESPONSE

Responses to reviewer's comments

Reviewer's comment: "My major concern with the manuscript is that not all discussions and conclusions are justified by the results and the authors' conclusions go beyond their primary objective. Some of my previous comments were not addressed adequately"

Responses to reviewer's comments

Thank you for the accurate comments about this issue, we fully agree with the reviewer's comments and suggestions. We have turned down discussion, conclusion and revised our manuscript as below listed.

Comment#1

"A study shows that certain ethnic groups do not meet current criteria of MetS until they have reached a more advanced degree of insulin resistance²²". Are the authors proposing to stop measuring MetS criteria in clinical practice, but instead to measure fasting insulin, or how do they vision this clinically? Their data demonstrate an association between fasting insulin and established MetS criteria; this does not demonstrate that fasting insulin can be used as an alternative for certain ethnic groups that do not meet formal MetS criteria.

Response#1

Thank you for the accurate comments about this issue, we agree with the reviewer's comments that our citation 22 is not appropriate. Thus we have deleted it throughout our manuscript.

(Page 4, line 79-80)

(Page 9, line 193-194)

(Page 11, line 235-237)

(Page 12, line 272-273)

In response to the reviewer, we do not propose to stop measuring metabolic syndrome (MetS) criteria in clinical practice. We only suggest that physicians be aware of the increased MetS risk with elevated fasting insulin (FI) levels. We have added this in our "Discussion":

"Elevated FI may act as an accompanying marker to enhance the risk of MetS. We do not propose to discard MetS criteria, but suggest that elevated FI may alert physicians on the risk of MetS in clinical settings of non-diabetic individuals."

(Page 11, line 239-241)

Comment#2

The authors show that fasting insulin >7.35 has some predictive power, but the positive and negative predictive values do not support the term "robust and reliable predictor", in my opinion.

Response#2

We fully agree with the reviewer's comments that our fasting insulin >7.35 is not a robust and reliable predictor. Thus, we have decided not to emphasize a FI cut-off value for the detection of MetS due to the FI range varies between different ethnic groups and laboratory assays. We have deleted some inadequate lines containing 7.35 throughout the manuscript.

(Page 10, line 219-222)

(Page 10, line 230-232)

(Page 11, line 244-246)

(Page 12, line 286-290)

Comment#3

The authors recommend lifestyle interventions for those who have fasting insulin >7.35, but - to my knowledge - there is no evidence to support this. I agree, however, that we are in need of large trials to determine if subjects with early stages of insulin resistance can benefit from interventions.

Response#3

Thanks for your accurate comments about the issue. We fully agree with the reviewer's suggestion. We have changed the words from "lifestyle modification should be considered when the FI level is > 7.35 $\mu\text{U/mL}$, due to the amelioration of metabolic abnormalities with diet or exercise interventions" to "healthy behavior should be considered when the FI level is relatively higher in the population. Though we are in need of large trials to determine if subjects with early stages of insulin resistance can benefit from interventions" (Page 11, line 243-247)

Comment#4

I think it should be made clear that any reported insulin cutoff value can only be applied to the insulin assay that was used. Due to large variation in insulin assays, the value of >7.35 should not be generalized to other laboratory sites.

Response#4

Thanks for your accurate comments about the issue. We have added this to our limitation: "Fourth, the FI cutoff value varies between different insulin assays, so physicians should be aware of this variation in clinical settings." (Page 11, line 255-257)

Comment#5

Their definition of diabetes should be included with the exclusion criteria/methods.

Response#5

We thank the reviewer's comments and suggestions. We added the following in the "Method":

Diabetes mellitus was defined as any of the followings

1. Previous diagnosis of diabetes mellitus
2. Recent use of oral anti-hyperglycemic drugs or insulin
3. Participants with fasting glucose $\geq 126\text{mg/dl}$

(Page 5, line 95-97)

We did not perform glucose tolerance test, this is a weak point of our study.